# Screening Severe Obstructive Sleep Apnea in Children with Snoring

**DOI:** 10.3390/diagnostics11071168

**Published:** 2021-06-26

**Authors:** Hui-Shan Hsieh, Chung-Jan Kang, Hai-Hua Chuang, Ming-Ying Zhuo, Guo-She Lee, Yu-Shu Huang, Li-Pang Chuang, Terry B.-J. Kuo, Cheryl C.-H. Yang, Li-Ang Lee, Hsueh-Yu Li

**Affiliations:** 1Sleep Center, Department of Otorhinolaryngology-Head and Neck Surgery, Chang Gung Memorial Hospital, Linkou Main Branch, Taoyuan 33305, Taiwan; hsieh1111@gmail.com (H.-S.H.); handneck@gmail.com (C.-J.K.); 2Department of Otolaryngology, Xiamen Chang Gung Hospital, Xiamen 361126, China; zhuomy@adm.cgmh.com.cn; 3College of Medicine, Chang Gung University, Taoyuan 33302, Taiwan; chhaihua@cgmh.org.tw (H.-H.C.); yushuhuang1212@gmail.com (Y.-S.H.); lpchuang1678@yahoo.com.tw (L.-P.C.); 4Department of Family Medicine, Chang Gung Memorial Hospital, Taipei Branch & Linkou Main Branch, Taoyuan 33305, Taiwan; 5Department of Industrial Engineering and Management, National Taipei University of Technology, Taipei 106344, Taiwan; 6Faculty of Medicine, School of Medicine, National Yang Ming Chiao Tung University, Taipei 11221, Taiwan; guosheli@gmail.com; 7Department of Otolaryngology, Taipei City Hospital, Ren-Ai Branch, Taipei 10629, Taiwan; 8Sleep Center, Department of Child Psychiatry, Chang Gung Memorial Hospital, Linkou Main Branch, Taoyuan 33305, Taiwan; 9Sleep Center, Department of Pulmonary and Critical Care Medicine, Chang Gung Memorial Hospital, Linkou Main Branch, Taoyuan 33305, Taiwan; 10Institute of Brain Science, National Yang Ming Chiao Tung University, Taipei 11221, Taiwan; tbjkuo@gmail.com (T.B.-J.K.); cchyang@gmail.com (C.C.-H.Y.)

**Keywords:** adenoidal-nasopharyngeal ratio, children, obstructive sleep apnea, oxygen desaturation index, snoring sound energy, tonsil size

## Abstract

Efficient screening for severe obstructive sleep apnea (OSA) is important for children with snoring before time-consuming standard polysomnography. This retrospective cross-sectional study aimed to compare clinical variables, home snoring sound analysis, and home sleep pulse oximetry on their predictive performance in screening severe OSA among children who habitually snored. Study 1 included 9 (23%) girls and 30 (77%) boys (median age of 9 years). Using univariate logistic regression models, 3% oxygen desaturation index (ODI3) ≥ 6.0 events/h, adenoidal-nasopharyngeal ratio (ANR) ≥ 0.78, tonsil size = 4, and snoring sound energy of 801–1000 Hz ≥ 22.0 dB significantly predicted severe OSA in descending order of odds ratio. Multivariate analysis showed that ODI3 ≥ 6.0 events/h independently predicted severe pediatric OSA. Among several predictive models, the combination of ODI3, tonsil size, and ANR more optimally screened for severe OSA with a sensitivity of 91% and a specificity of 94%. In Study 2 (27 (27%) girls and 73 (73%) boys; median age, 7 years), this model was externally validated to predict severe OSA with an accuracy of 76%. Our results suggested that home sleep pulse oximetry, combined with ANR, can screen for severe OSA more optimally than ANR and tonsil size among children with snoring.

## 1. Introduction

Obstructive sleep apnea (OSA) is characterized by intermittent episodes of upper airway collapse during sleep. It is a chronic and severe breathing-related disorder, which affects approximately 2 to 3% of school-aged children [1,2]. A substantial volume of evidence has linked pediatric OSA and cardiovascular/metabolic dysfunctions [1,2]. Some research has also suggested a connection between pediatric OSA and neurocognitive/behavioral impairment, but the results are still inconsistent and inconclusive [3]. Adenotonsillectomy remains the preferred treatment for pediatric OSA [4,5]; however, older children have been reported to have less satisfactory surgical outcomes [6]. Therefore, delay in the diagnosis of severe pediatric OSA has been one of the major attributable causes for compromised treatment outcomes for pediatric sleep-disordered breathing.

Snoring is a hallmark of pediatric OSA, and around 20% of children who are frequent and loud snorers have OSA [7]. However, there are only a few studies using objective evaluations of snoring sound for the diagnosis of pediatric OSA [8,9]. Standard in-laboratory nocturnal polysomnography is the best method to diagnose OSA; however, it is time-consuming and labor-intensive, and hence the waiting list is usually very long [10]. Furthermore, the high cost and poor tolerability of the polysomnography apparatus can also delay a prompt diagnosis and treatment in children with OSA [11,12]. An increasing number of rapid screening tools have been developed to better stratify children with snoring for their needs of polysomnography and treatment.

For example, the OSA-18 questionnaire has been used to screen for pediatric OSA [13]. Additionally, adenoid and tonsillar hypertrophy, traditional risk factors for pediatric OSA, have been used to predict disease severity [14]. Nocturnal pulse oximetry, a typical home sleep apnea test, has shown excellent ability to identify pediatric OSA [15]. Recently, snoring sound analysis, an objective evaluation of snoring sounds has been demonstrated to have potential in screening for OSA [9,16]. Even though the pathophysiology behind these rapid screening tests is distinctly different, they can all be used to predict the severity of OSA. The utility and limitations of anatomy-based classification systems and snoring sound analysis have been elucidated in some previous research. Nevertheless, to the best of our knowledge, no previous study has compared the diagnostic ability of these screening tools simultaneously in the same pediatric cohort [9,17,18]. It is also a novel approach to develop various combinations of the aforementioned tests and report their performance in identifying severe OSA.

We hypothesized that clinical variables, home snoring sound analysis, and home sleep pulse oximetry had different performances in screening severe OSA among children with habitual snoring. The first aim of the study was to evaluate the prediction performance for severe pediatric OSA of the aforementioned various measures, including home snoring sound analysis and home sleep pulse oximetry. The second aim of the study was to develop a combined predictive model for severe pediatric OSA with measurements from these existing screening tools to facilitate timely diagnosis and early intervention.

## 2. Materials and Methods

### 2.1. Ethical Considerations 

This retrospective cross-sectional study was approved by the Institutional Review Board of the Chang Gung Memorial Foundation (No. 202000873B0). The study was conducted according to the guidelines of the Declaration of Helsinki. The requirement for written informed consent was waived. 

### 2.2. Participants

We reviewed medical records of 150 children with OSA at the Department of Otorhinolaryngology-Head and Neck Surgery at Chang Gung Memorial Hospital, Linkou Main Branch (Taoyuan, Taiwan) between 1 March 2015, and 31 Jun 2019. 

The investigation comprised two parts: Study 1 (discovery study) and Study 2 (validation study).

#### 2.2.1. Study 1: Development of a Combined Model for Screening Severe Pediatric OSA 

Data obtained from the first 42 consecutive subjects with habitual snoring (snoring three or more times per week) were used to compare the predictive performance of clinical variables, home snoring sound analysis, and home sleep pulse oximetry in screening severe pediatric OSA. A combined predictive model was further developed with measures from the abovementioned existing screening tools.

All 42 patients had a standard in-laboratory polysomnography before home sleep apnea tests [9]. A case-matched control design was adopted to reduce the confounding effects from variables known to be associated with OSA severity [19]. There were 23 children with severe OSA and 19 age-, sex-, and BMI-matched control children with non-severe OSA.

#### 2.2.2. Study 2: External Validation of the Combined Model for Screening Severe Pediatric OSA

The remaining 108 consecutive subjects with snoring were included in the external validation of the combined model. Each of them underwent nocturnal pulse oximetry at home and subsequently received one standard in-laboratory polysomnography to confirm the diagnosis and severity of OSA.

The inclusion criteria were: (a) age 5–12 years; (b) subjective symptoms of snoring (more than once a week on most nights [20]) and at least one major symptom or sign of OSA (such as increased effort to breathe, witnessed apnea episodes, restless sleep, diaphoresis, frequent awakening, or enuresis) [21]; and (c) having data of in-laboratory polysomnography and nocturnal pulse oximetry or snoring sound recordings. The exclusion criteria were: (a) complicated co-morbidities (such as craniofacial anomalies or neuromuscular disorders); (b) complex conditions that profoundly disturbed snoring sound collection (such as sleepwalking and bruxism) [22]; and (c) insufficient data of pulse oximetry and snoring sound analysis. 

### 2.3. Polysomnography

All 150 patients underwent standard in-laboratory full-night polysomnography (Nicolet Biomedical Inc., Madison, WI, USA) with simultaneous video recording to document OSA parameters [9,22]. Oronasal airflow was measured by a thermistor and oxyhemoglobin saturation was recorded using a finger pulse oximeter (model 340; Palco Laboratories, Santa Cruz, CA, USA). The oximeters employed a sampling frequency of 0.5 Hz. The same registered sleep technologists performed polysomnograms, and the study investigators (Huang, Y.-S. and Chuang, L.-P.) validated polysomnography manually to ensure the quality of interpretation.

The apnea–hypopnea index (AHI) was defined as the sum of all apneas (≥90% decrease in airflow for a duration of ≥2 breaths) plus hypopneas (≥50% decrease in airflow and either ≥3% desaturation or electroencephalographic arousal, for a duration of ≥2 breaths), divided by the number of hours of total sleep time, according to the 2012 American Academy of Sleep Medicine Scoring Manual [22]. The obstructive AHI, apnea index (AI), the number of SpO_2_ drops ≥ 3% per hour of recording (ODI3), and snoring index (SI) were recorded for further comparisons. In the present study, the diagnosis of OSA was defined by an obstructive AHI ≥ 2.0 events/h or an obstructive AI ≥ 1.0 events/h [23,24]. Furthermore, the patients were categorized as having severe (obstructive AHI ≥ 10.0 events/h) or non-severe (obstructive AHI < 10.0 events/h) OSA [25].

### 2.4. Clinical Variables

The 42 parents in Study 1 were asked to complete the Chinese version of the OSA-18 questionnaire to evaluate their quality of life [13]. The OSA-18 questionnaire consists of 18 items that were scored on a 7-point ordinal scale (overall range, 18‒126), and has been shown to have excellent test–retest reliability [26].

We directly inspected the tonsils and graded the tonsil size on a scale from 0–4 for each patient using the Brodsky grading scale [27], which has been shown to have excellent interobserver and intraobserver reliability [28]. We also measured the adenoidal-nasopharyngeal ratio (ANR) to grade the adenoid size using lateral radiography of the nasopharynx [29]. For minimizing the possible differences between operators, two senior investigators (Lee, L.-A. and Li, H.-Y.) measured the tonsil size and ANR following the procedure previously used [30].

### 2.5. Home Sleep Apnea Tests

Unattended nocturnal SpO_2_ and snoring signals were recorded in a home environment. The caregivers were shown how to simultaneously start a wearable wrist pulse oximeter (3100 WristOx, Nonin Medical, Inc., Minneapolis, MN, USA). The fingertip sensor was used to obtain the SpO_2_ signal (The Novametrix 520 and the 3100 WristOX) and was configured at a 0.5 Hz sampling frequency. Furthermore, the portable sound recorder using linear pulse-code modulation (PCM-D50, Sony Electronics Inc., Tokyo, Japan) was positioned 50 cm above the child’s head when they fell asleep between 21:00 to 24:00 [9].

#### 2.5.1. Pulse Oximetry

In the preparation phase, we found that the children frequently disconnected the fingertip sensor of the pulse oximeter during sleep, even though we fixed the sensor with stick tapes. Therefore, we reviewed the wearing efficiency to ensure the quality of PO measurements. Seven of ten subjects had ≥60% valid signal data (i.e., available pulse oximetry data after removing motion artifacts and awakenings) for full-night recordings (range, 6–10 h), whereas nine had ≥60% valid signal data for the first 150 min of recordings. Furthermore, the first snoring signal frequently occurred later than approximately 30 min after starting recording. For a more specific analysis of SpO_2_ during sleep, we analyzed consecutive 90 min of pulse oximetry data after the first snoring signal to include at least one sleep cycle [31,32]. 

The SpO_2_ was automatically recorded every 1 s and stored in memory during the full-night sleep. The ODI3 was provided by the software (nVISION 6.0e, Nonin Medical, Inc., Plymouth, MN, USA). All 150 patients underwent nocturnal pulse oximetry measurements. Cases with insufficient data of pulse oximetry (<75 min of pulse oximetry data after removal of motion artifacts and/or awakenings [33]) were excluded from statistical analysis. 

#### 2.5.2. Snoring Sound Analysis 

We recorded the full-night snoring signals of the first 42 children using the same protocol as described in detail in our previous study [9,34]. Briefly, at least 6 h of snoring sounds were recorded at a sampling rate of 44,100 Hz with a 16-bit A/D converter. The sound signals were first reviewed by a playback system to identify the first snore and manually removed all of the high-frequency sounds (>1000 Hz) from acoustic analysis to reduce the wall and/or ceiling reactions [35]. It has been acknowledged that it is not straightforward to differentiate the patient sounds and the surrounding noise from environmental sounds [36]. Thus, the 1 min signals before the first snore were considered background noise, and the root mean square was obtained as a baseline to distinguish snores from noise. 

Snore epochs were considered as the signals of an analytic window with energy at least 6 dB stronger than background noise; all others were considered to be noise epochs and snores were considered of consecutive snore epochs lasting 0.5–3 s [37], thus the snore index (SI, events/min) was calculated. The power spectrum was acquired for each window using fast Fourier transformation (41 Hz to 1000 Hz) with a frequency resolution of 4 Hz. The energy levels of various frequency domains (in dB), denominated snoring sound energy (SSE, dB), were acquired using a specially developed software program (LabVIEW, National Instruments Corp., Austin, TX, USA). Snoring sound analysis, including the SI and SSE, was performed for each record parallel to the pulse oximetry time (time window, 0.25 s; snore epochs, ≥2 folds of the baseline root mean square; epoch duration, 0.5–3 s). Cases with insufficient data of snoring sound analysis (<300 min of pulse oximetry data after removal of noise and/or awakenings) were excluded from statistical analysis.

### 2.6. Statistical Analysis 

The D’Agostino and Pearson normality test showed that most of the variables were non-normally distributed, and the descriptive statistics of these variables were presented as the median and interquartile range (IQR). Differences in the variables between groups were analyzed using the Mann–Whitney *U* test for comparisons between non-severe OSA and severe OSA subgroups and the Wilcoxon signed-rank test for comparisons of related samples. Differences in categorical variables between non-severe OSA and severe OSA subgroups were analyzed using Fisher’s exact test. Effect sizes were calculated with the use of Hodges–Lehmann method for Mann–Whitney *U* test, or an odds ratio calculation for Fisher’s exact test. Associations among AHI and other variables of interest were evaluated using the Spearman correlation test. The variables of interest were further dichotomized according to the optimal cut-off value using receiver operating curves with the Youden’s index [38]. The dichotomized variables were then assessed using univariate and multivariate logistic regression models. For selecting optimal variables and improving predictive performance, multivariate categorical regression models of all dichotomized variables with the logistic least absolute shrinkage and selection operator (LASSO) and bootstrap resampling (*n* of samples = 50; 100 runs) were performed [39]. All *p*-values were two-sided, and statistical significance was accepted at *p* < 0.05. All statistical analyses were performed using Graph Pad Prism 9.0 for Windows (Graph Pad Software Inc., San Diego, CA, USA) and SPSS 25.0 statistical package for Windows (International Business Machines Corp., Armonk, NY, USA).

## 3. Results

### 3.1. Study 1: Development of a Combined Model for Screening Severe Pediatric OSA

#### 3.1.1. Case Diagram

Figure 1 demonstrates the case diagram. In Study 1, three children were excluded from statistical analysis due to insufficient pulse oximetry data. Therefore, a total of 39 Taiwanese children of Han ancestry with habitual snoring and OSA (nine (23%) girls and 30 (77%) boys; median age, 9 (IQR: 6–10) years; median body mass index (BMI) z-score of 1.2 (IQR: 0.2–1.9)) were included for further analysis. 

#### 3.1.2. Difference in Patients Characteristics of Study 1

In Study 1, the median AHI was 13.8 events/h (IQR: 3.2–25.3), including 21 (54%) with severe OSA and 18 (46%) with non-severe OSA. The characteristics of these two subgroups categorized by OSA severity are shown in Table 1. The severe OSA subgroup had a significantly larger tonsil size (effect size = 1; 95% confidence interval (CI): 0–1), higher ANR (effect size = 0.12; 95% CI: 0.06–0.19), higher obstructive AHI (effect size = 19.0; 95% CI: 13.8–27.8), ODI3 (effect size = 18.9; 95% CI: 14.1–25.7), and SI (effect size = 160.0; 95% CI: 56.0–295.0), compared to the non-severe OSA subgroup. There were no significant differences in age, sex, BMI z-score, OSA-18, and obstructive AI.

#### 3.1.3. Differences in Home Sleep Apnea Tests of Study 1

Table 2 shows the 90 min pulse oximetry and 360 min snoring sound analysis variables of Study 1 measured at home in the overall cohort as well as both severe and non-severe OSA subgroups. The home-ODI3 was significantly related to the polysomnography-ODI3 (*r* = 0.80, *p* < 0.001) using the same plethysmograph technology. However, home-SI was not statistically significantly associated with the polysomnography-SI (*r* = 0.24, *p* = 0.15) with the difference in the given nature of measurements (snoring sound analysis: SSE; polysomnography: snoring vibration energy).

The median duration of polysomnography and home sleep apnea tests was 4.7 (interquartile range: 3.5–5.3) weeks. The severe OSA subgroup had a significantly higher ODI3 compared to the non-severe OSA subgroup in both subgroups (effect size = 7.3; 95% CI: 3.5–20.1).

Using the 6 h snoring sound analysis, all patients had available data ≥90% to analyze the measurements taken at home. The severe OSA subgroup had a significantly higher SSE of 801–1000 Hz than the non-severe subgroup (effect size = 9; 95% CI: 0–20.3). The differences in other snoring sound analysis variables between the subgroups were not significant.

#### 3.1.4. Associations between Variables Related to the AHI of Study 1

As expected, the associations among AHI and tonsil size (*r* = 0.41, *p* = 0.01), ANR (*r* = 0.41, *p* = 0.01), ODI3 (*r* = 0.84, *p* < 0.001), and SSE of 801–1000 Hz (*r* = 0.39, *p* = 0.01) were significant. The OSA-18 was significantly associated with tonsil size (*r* = 0.39, *p* = 0.01), and SSE of 801–1000 Hz (r = 0.39, *p* = 0.01). Tonsil size was also significantly related to SSE of 801–1000 Hz (r = 0.38, *p* = 0.01). Finally, ANR was also related to ODI3 (*r* = 0.46, *p* = 0.002). 

#### 3.1.5. Predictors and Prediction Models for Severe OSA of Study 1

In Study 1, we examined receiver operating curves for the prediction of severe OSA using single patient characteristics or home sleep apnea test variables (Table 3). The best performers included ODI3 ≥ 6.0 events/h, ANR ≥ 0.78, tonsil size = 4, SSE of 801–1000 Hz ≥ 22.0 dB, and OSA-18 ≥ 77, in descending order of odds ratio.

Using multivariate logistic regression analysis, we constructed several prediction models, including an anatomical model and a home sleep apnea test model (Table 4). Using multivariate categorical regression analysis with the LASSO (Figure 2), a combined predictive model was constructed. 

The anatomical model included ANR ≥ 0.78 and tonsil size = 4 as independent predictors for severe OSA. Accordingly, the anatomical model achieved an area under the receiver operating characteristic curve (AUC) of 0.81 (*p* = 0.001). The sensitivity, specificity, positive predictive value (PPV), and negative predictive value (NPV) of a cut-off value of one risk factor were 100%, 61%, 75%, and 100%, respectively. The accuracy of the anatomical model was 82%.

The home sleep apnea test model included ODI3 ≥ 6.0 events/h and SSE of 801–1000 Hz ≥ 22.0 dB as independent variables for severe OSA. Accordingly, the home sleep apnea test model achieved an AUC of 0.81 (≥1 predictor; *p* = 0.001) with a sensitivity of 90%, a specificity of 72%, PPV of 79%, NPV of 87%, and accuracy of 82%, respectively.

The combined model included ODI3 ≥ 6.0 events/h, tonsil size = 4, and ANR ≥ 0.78 as predictors for severe OSA. For weighting the regression coefficient, (the presence of ODI3 ≥ 6.0 events/h) × 3 + (the presence of tonsil size = 4) × 2 + (the presence of ANR ≥ 0.78) × 1. Therefore, the combined model achieved an AUC of 0.93 (≥4 scores; *p* < 0.001) with a sensitivity of 91% and a specificity of 94%. The PPV was 95%, and the NPV was 89% in this model with the highest accuracy of 92%.

### 3.2. Study 2: External Validation of the Combined Model for Screening Severe Pediatric OSA

#### 3.2.1. Case Diagram

In Study 2, one patient had been diagnosed with asthma, and seven children with insufficient pulse oximetry data were excluded from the validation study (Figure 1). A total of 100 children with OSA (29 (29%) girls and 71 (71%) boys) with a median age of 7 (IQR: 6–10) years and a median BMI z-score of 0.6 (IQR: −0.6–1.8) were included for analysis. The others had no related lung or heart comorbidities that may cause desaturation.

#### 3.2.2. Difference in Patients Characteristics of Study 2

The median AHI was 9.6 events/h (IQR: 5.2–22.8), including 47 (47%) with severe OSA and 53 (53%) with non-severe OSA. The characteristics of these two subgroups categorized by OSA severity are shown in Table 5. The severe OSA subgroup had a significantly higher ANR (effect size = 0.07; 95% CI: 0.03–0.14), higher obstructive AHI (effect size = 18.4; 95% CI: 14.4–24.3), and higher obstructive AI (effect size = 3.7; 95% CI: 2.2–6.5) compared to the non-severe OSA subgroup. Furthermore, the median age of the validation cohort was significantly lower than that of the discovery cohort (effect size = −1; 95% CI: −2–0; *p* = 0.02), whereas the distributions of sex, BMI z-score, tonsil size, ANR, AHI, and AI of both cohorts were compatible (all *p* > 0.05).

#### 3.2.3. Home Sleep Apnea Test Variables of Study 2

The 90 min pulse oximetry of Study 2 was also measured during sleep at home without staff supervision in the severe and non-severe OSA subgroups. The median duration of home pulse oximetry and polysomnography was 11.4 (interquartile range: 6.8–16.0) weeks. The median ODI3 was 4.9 events/h (IQR: 1.9–16.5). The severe OSA subgroup had a significantly higher ODI3 (median, 21.3; IQR: 11.3–40.8) compared to the non-severe OSA subgroup (median, 4.3; IQR: 2.3–7.9) in both cohorts (effect size = 16.9; 95% CI: 12.3–20.4).

#### 3.2.4. Validation of the Predictive Models from Study 1 in the Cohort of Study 2

The prediction accuracy of the anatomical model was 61% and the Kappa was 0.22 (*p* = 0.01) (Table 6). The prediction accuracy of the combined model was 76% and the Kappa was 0.56 (*p* < 0.001).

## 4. Discussion

Home sleep apnea tests with technically adequate devices have been used to diagnose moderate to severe OSA in uncomplicated adult patients [40] and school-aged children [41,42] with sleep-disturbed breathing; however, these devices are not recommended for the diagnosis of pediatric OSA [43]. Home sleep apnea tests have fewer first-night effects compared to overnight in-lab polysomnography [44]. A familiar environment and simple devices may minimize the interference on one’s usual sleep and also improve children’s compliance [45]. A combination of simplified cardiorespiratory montage and video recording have been proven to accurately evaluate OSA in pediatric patients with adenotonsillar hypertrophy [46]. In this study, the ODI3 might provide greater accuracy than the SSE of 801–1000 Hz did.

Home sleep pulse oximetry is a promising tool to screen for OSA in symptomatic children, primarily due to the significant difference in ODI3 between children with and without OSA [47,48,49,50]. Brouillette reported that children identified by pulse oximetry testing were likely to have clinically significant disease severity [51]. Furthermore, using home sleep apnea tests to estimate the severity of OSA could assist clinicians with timely referrals in areas with limited access to polysomnography and reduce the workload of sleep technicians. However, easy dislodgement of the finger sensor is a major disadvantage of unattended pulse oximetry. Approximately 10% to 20% of hospital and home sleep pulse oximetry failed to provide technically acceptable data due to poor contact between the sensor and the child’s finger [33,42,52]. Therefore, Bhattacharjee commented that the failure rate of home pulse oximetry and cardiorespiratory measures was relatively high for studies that were attended by night technologists and for older children [53].

In this study, we provided a feasible protocol to collect valid SpO_2_ signals to assess disease severity. The majority of children (139/149 (93%)) accomplished the first 90-min recordings of SpO_2_ at home. Notably, the first sleep cycle is usually a slow-wave sleep with the smallest number of apneas [54]. However, those with significant oxygen desaturations during this sleep stage could be assumed to be at risk of severe OSA. Additionally, parents are more likely to keep vigilance during this early night sleeping period, assist the measurement, and re-connect the pulse oximetry device when dislodged.

Previous studies have adopted different cut-off values of ODI3 for screening pediatric OSA [55,56,57,58]. The diversity could be caused by varied sample size and screening of the different OSA severity in each study. An ODI3 cut-off value of 5 events/h was reported to identify children with moderate abnormalities [56], while the cut-off value of 25 events/h was defined to diagnose severe pediatric OSA [58]. The optimal ODI3 corresponding to the polysomnography-defined severe OSA was established as ≥6.0 events/h in our discovery study and demonstrated a sensitivity of 90% and a specificity of 83%. This discrepancy might be due to the difference in ODI3 between the early night and the full night. However, technically unacceptable data of the full night sleep limit the accuracy of SpO_2_ analysis, and artificial intelligence may help to precisely remove motion artifacts, poor-quality signals, and awakenings [59].

Furthermore, Hornero reported that home sleep pulse oximetry with an automated neural network algorithm accurately detected severe OSA with an AUC of 0.91 [15]. Therefore, single-channel nocturnal pulse oximetry may be an alternative to respiratory polygraphy and polysomnography in an unattended setting to screen for OSA among children [15,50]. In this study, we used a commercially available pulse oximetry system to assess SpO_2_, improved the validity and efficacy of this test, and found that ODI3 during a 90 min period after the first snoring signal could predict severe OSA with an AUC of 0.87. Additionally, ODI3 performed better than the other independent predictors in the detection of severe OSA. For example, ODI3 detected severe OSA with greater sensitivity (90% and 86%, respectively) and specificity (83% and 67%, respectively) than ANR. A recent study also reported that pulse oximetry provided satisfactory diagnostic performances in detecting moderate and severe OSA [60]. Our findings further support the use of home sleep pulse oximetry to screen for severe OSA among children with snoring.

In another home sleep apnea test, we used a non-contact technology to detect snoring sounds to avoid the problem of sensor dislodgement and therefore classified the severity of OSA more efficiently. An elevated SSE of 850–2000 Hz has been reported to be a warning sign of severe OSA in adults [37]. Furthermore, an increased SSE of 801–1000 Hz has been shown to predict more severe OSA (AHI > 15 events/h) and to indicate multiple-level obstructions due to a low surgical success rate in children with OSA [9]. In this study, the accuracy of SSE of 801–1000 Hz ≥ 22.0 dB was poorer than that of ODI3 ≥ 6.0 events/h for predicting severe OSA (67% versus 87%).

Clinicians usually evaluate the risk of severe OSA for children with clinical history, physical examination, and questionnaire-based tools. However, these approaches are with limited sensitivity and specificity [61]. In the present study, differences in age, sex, BMI z-score, and OSA-18 between subgroups in the discovery cohort could not differentiate severe OSA from non-severe OSA at the first evaluation. Both the ANR and tonsil size could provide a more reliable prediction of severe pediatric OSA than the OSA-18 instrument. The sensitivity of OSA-18 to predict severe OSA was 57% in our study, which was compatible with the pooled sensitivity of a recent meta-analysis [62]. Our result further supports that the predictive value of the OSA-18 for severe OSA is weak [63].

The optimal cut-off value of the ANR was 0.78 for predicting severe OSA with the largest AUC comparing with other single predictors. The ANR is significantly associated with age [64] and AHI [19]. Clinically, an ANR ≥ 0.73 may be considered indicative of pathological enlargement of the adenoids [65]. Accordingly, our results strengthened the utility of the ANR to screen severe OSA in children with snoring. Furthermore, the anatomical model (either ANR ≥ 0.78 or tonsil size = 4) predicted severe OSA more precisely than the single anatomic predictors.

This study provides a practical perspective for clinicians to predict severe pediatric OSA more efficiently. In children with habitual snoring, we suggest using the anatomical model or the combined model to screen for severe OSA since they both have acceptable accuracy (82% and 92%, respectively). However, the predictive performances of both the anatomical and combined modules obviously reduced in the validation cohort. These results might be because of the younger age of the validation cohort, which reduced the predictive performance of the ANR and tonsil size (Table 6). Nevertheless, our simple predictive models may assist clinicians to better stratify at-risk children for severe OSA and facilitate a timely diagnosis and early intervention. This proactive management could reduce OSA-related health costs in the future.

Our study had some limitations. First, the study was retrospective cross-sectional and had a small sample size. Second, despite the fact that variables obtained from nocturnal pulse oximetry were distinguishable from those from polysomnography in terms of testing environments and inclusion of respiratory events, ODI3 might be partially autocorrelated with the AHI since oxygen desaturation was part of the definition of hypopneas. Third, an adaption night standard polysomnography was not included in our protocol. Although an adaption night is considered necessary only for nocturnal respiratory pattern investigations, the first-night effect may still influence the sleep architectures in subjects [66]. Fourth, selection bias that might have existed concerning the single ethnicity, a predominance of boys, a subgroup of children with snoring, a confined age spectrum of 5–12 years, and a lack of a normal control group. Nevertheless, the combined model reached a considerable accuracy in the subsequent Study 2 cohort. Further prospective investigation with a sample larger in size and more heterogenous in demographics (e.g., different ethnic groups) is warranted to verify the findings of this research.

## 5. Conclusions

Increased tonsil size, ANR, ODI3, and SSE of 801–1000 Hz were related to severe pediatric OSA. We found that ODI3 had a better performance to predict severe OSA than the other variables of interest in children with loud snoring. Therefore, non-invasive, inexpensive pulse oximetry can be widely used to screen for severe pediatric OSA in an unattended home setting with acceptable sensitivity and specificity. Furthermore, the combination of ODI3 with the tonsil size and ANR might perform better in predicting severe pediatric OSA.

## Figures and Tables

**Figure 1 diagnostics-11-01168-f001:**
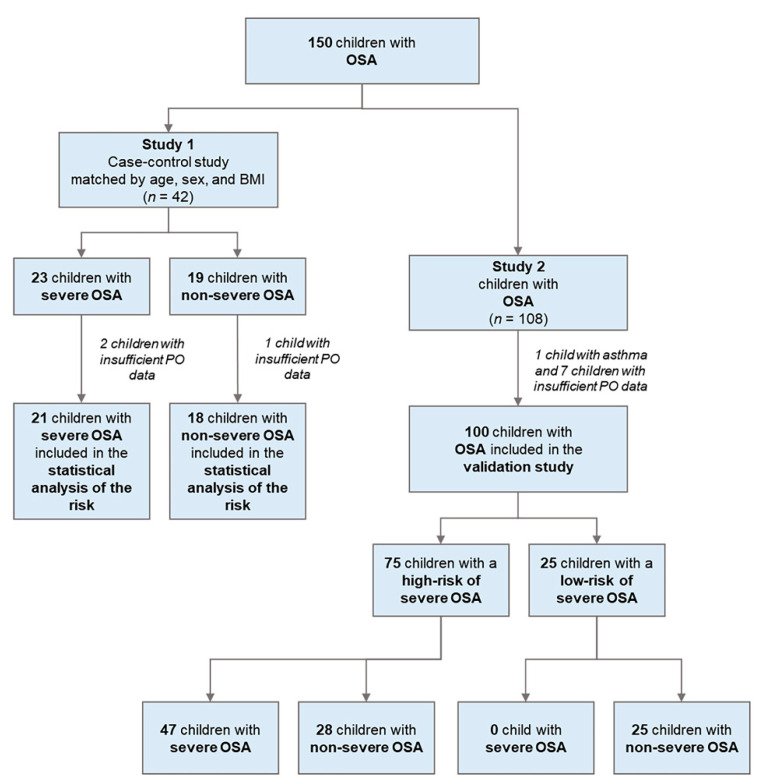
Case diagram. We included 150 eligible children with obstructive sleep apnea (OSA) for reviewing and excluded 11 patients from statistical analysis due to insufficient pulse oximetry (PO) data or a history of asthma. Twenty-one children with habitual snoring and severe OSA and 18 age-, sex-, and body mass index (BMI)-matched control children with habitual snoring and non-severe OSA comprised the final Study 1. One hundred additional children with OSA formed the final Study 2. Abbreviation: *n*: number.

**Figure 2 diagnostics-11-01168-f002:**
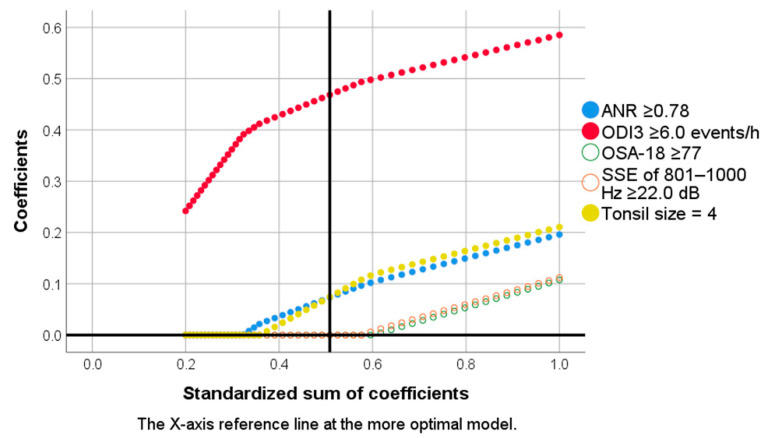
Logistic least absolute shrinkage and selection operator paths for severe pediatric obstructive sleep apnea (OSA). Five variables, including 3% oxygen desaturation index (ODI3) ≥ 6.0 events/h, adenoid/nasopharynx ratio (ANR) ≥ 0.78, tonsil size—4, snoring sound energy (SSE) of 801–1000 Hz ≥ 22.0 dB, and OSA-18 ≥ 77, were analyzed using the multivariate categorical regression models. The X-axis reference line indicates the combined model (three variables).

**Table 1 diagnostics-11-01168-t001:** Patients’ characteristics of Study 1.

Characteristics	Overall	Severe OSA	Non-Severe OSA	Effect Size, Median of Difference (95% CI) or Odds Ratio (95% CI) ^1^	*p*-Value ^2^
	(*n* = 39)	(*n* = 21)	(*n* = 18)		
Demographic data					
Age (y)	9 (6–10)	9 (6–10)	9 (7–12)	0 (−1–2)	0.43
Sex (female/male)	9/30	3/18	6/12	0.3 (0.1–1.6)	0.26
BMI (kg/m^2^) z-score	1.2 (0.2–1.9)	1.4 (0.8–2.0)	1.0 (−0.2–1.5)	0.5 (−0.2–1.2)	0.20
Subjective symptomatic data					
OSA-18 (18–126)	75 (60–84)	77 (65–84)	70 (58–83)	8 (−5–19)	0.10
Anatomical data					
Tonsil size (1‒4)	3 (3–4)	4 (3–4)	3 (2–3)	1 (0–1)	**<0.01**
ANR	0.80 (0.70–0.86)	0.84 (0.80–0.92)	0.72 (0.63–0.81)	0.12 (0.06–0.19)	**<0.001**
Polysomnographic data					
Obstructive AHI (events/h)	13.8 (3.2–25.3)	21.6 (16.9–34.6)	3.0 (1.7–5.8)	19.0 (13.8–27.8)	**<0.001**
Obstructive AI (events/h)	1.8 (0.6–3.8)	3.3 (0.9–6.7)	1.0 (0.5–2.1)	1.4 (0–3.3)	0.053
ODI3 (events/h)	11.7 (2.8–24.0)	22.2 (16.7–33.9)	2.8 (1.6–5.7)	18.9 (14.1–25.7)	**<0.001**
SI (events/h)	180.0 (99.1–391.1)	335.9 (170.3–459.1)	114.7 (38.0–208.5)	160.0 (56.0–295.0)	**0.003**

Data are summarized as median and interquartile range or number. Abbreviations: AHI: apnea–hypopnea index; AI: apnea index; ANR: adenoid-nasopharynx ratio; BMI: body mass index; ODI3: 3% oxygen desaturation index; OSA: obstructive sleep apnea; SI: snoring index. ^1^ Effect sizes were calculated with the use of Hodges–Lehmann method for Mann–Whitney *U* test, or odds ratio calculation for Fisher exact test. ^2^ Data were compared between the severe OSA subgroup and non-severe OSA subgroup using the Mann–Whitney *U* test or Fisher’s exact test as appropriate. Significant *p*-values are marked in bold.

**Table 2 diagnostics-11-01168-t002:** Home sleep apnea test variables of Study 1.

Characteristics	Overall	Severe OSA	Non-Severe OSA	Effect Size, Median of Difference (95% CI) or Odds Ratio (95% CI) ^1^	*p*-Value ^2^
Patients	39	21	18		
Data of pulse oximetry					
ODI3 (events/h)	6.7 (4.7–12.0)	11.5 (7.5–29.0)	4.7 (3.5–5.3)	7.3 (3.5–20.1)	**<0.001**
Data of snoring sound analysis					
SI (events/h)	752 (236–1219)	929 (348–1261)	399 (192–1101)	194 (−115–652)	0.23
SSE of 21–200 Hz (dB)	38.0 (22.4–53.5)	44.7 (20.8–53.5)	35.7 (29.6–48.7)	3 (−20–17)	0.73
SSE of 201–400 Hz (dB)	31.0 (15.0–40.2)	34.4 (15.8–40.7)	27.9 (13.1–34.9)	6 (−7–17)	0.29
SSE of 401–600 Hz (dB)	18.2 (3.5–35.7)	13.4 (7.7–41.2)	18.8 (0–33.8)	3 (−10–13)	0.61
SSE of 601–800 Hz (dB)	14.9 (0–27.0)	15.0 (1.1–28.2)	14.4 (0–24.2)	1 (−7–14)	0.41
SSE of 801–1000 Hz (dB)	11.4 (0–23.2)	15.9 (4.8–29.0)	4.4 (0–14.8)	9 (0–20.3)	**0.03**

Data are summarized as median and interquartile range. Abbreviations: ODI3: 3% oxygen desaturation index; OSA: obstructive sleep apnea; SI: snoring index; SSE: snoring sound energy. ^1^ Effect sizes were calculated with the use of Hodges–Lehmann method for Mann–Whitney *U* test, or odds ratio calculation for Fisher exact test. ^2^ Data were compared between severe OSA subgroup and non-severe OSA subgroup using the Mann–Whitney *U* test. Significant *p*-values are marked in bold.

**Table 3 diagnostics-11-01168-t003:** Clinical variables as potential predictors of severe OSA in Study 1.

	Receiver Operating Characteristic Curves				Binary Logistic Regression Models			Predictive Performance				
**Predictors**	**Cut-Off Value**	**AUC**	**95% CI**	***p*-Value**	**OR**	**95% CI**	***p*** **-Value**	**Sen**	**Spec**	**PPV**	**NPV**	**Acc**
ODI3	≥6.0 events/h	0.87	0.74–0.99	**<0.001**	47.5	7.0–321.7	**<0.001**	90%	83%	86%	88%	87%
ANR	≥0.78	0.76	0.60–0.92	**<0.01**	12.0	2.5–57.5	**0.002**	86%	67%	75%	80%	77%
Tonsil size	4	0.68	0.51–0.85	0.06	10.0	1.1–93.4	**0.04**	52%	83%	79%	60%	67%
SSE of 801–1000 Hz	≥22.0 dB	0.68	0.51–0.85	0.05	7.3	1.3–39.9	**0.02**	48%	89%	83%	59%	67%
OSA-18	≥77	0.62	0.44–0.80	0.21	2.7	0.7–9.9	0.14	57%	67%	67%	57%	62%

Abbreviations: Acc: accuracy; ANR: adenoid-nasopharynx ratio; AUC: area under the curve; CI: confidence interval; PPV: positive predictive value; NPV: negative predictive value; ODI3: 3% oxygen desaturation index; OR: odds ratio; OSA: obstructive sleep apnea; Sen: sensitivity; Spec: specificity; SSE: snoring sound energy. Significant *p*-values are marked in bold.

**Table 4 diagnostics-11-01168-t004:** Prediction models of severe OSA of Study 1.

	Binary Logistic Regression Models			Receiver Operating Characteristic Curves				Predictive Performance				
Predictors	OR	95% CI	*p*-Value	Cut-Off Value	AUC	95% CI	*p*-Value	Sen	Spec	PPV	NPV	Acc
The anatomical model				≥1 risk factor	0.81	0.66–0.96	**0.001**	100%	61%	75%	100%	82%
ANR ≥ 0.78	13.9	2.4–81.6	**0.004**									
Tonsil size = 4	6.7	1.1–43.1	**0.04**									
The home sleep apnea test model				≥1 risk factor	0.81	0.67–0.96	**0.001**	90%	72%	79%	87%	82%
ODI3 ≥ 6.0 events/h	38.3	5.4–270.3	**<0.001**									
SSE of 801–1000 Hz ≥ 22.0 dB	4.0	0.4–39.0	0.23									
The combined model				Total score = (ODI3 ≥ 6.0 events/h) × 3 + (Tonsil size = 4) × 2 + (ANR ≥ 0.78) × 1								
	**Beta**	**SE**	***p*** **-value**	**Cut-off value**	**AUC**	**95% CI**	***p*** **-** **V** **alue**	**Sen**	**S** **pec**	**PPV**	**NPV**	**Acc**
ODI3 ≥ 6.0 events/h	0.61	0.15	**<0.001**	≥4	0.93	0.83–1.00	**<0.001**	91%	94%	95%	89%	92%
Tonsil size = 4	0.21	0.13	**0.** **03**									
ANR ≥ 0.78	0.27	0.11	0.13									

Abbreviations: ACC: accuracy; ANR: adenoid/nasopharynx ratio; AUC: area under the curve; CI: confidence interval; NPV: negative predictive value; ODI3: 3% oxygen desaturation index; OR: odds ratio; PPV: positive predictive value; SE: standard error; Sen: sensitivity; Spec: specificity; SSE: snoring sound energy. Significant *p*-values are marked in bold.

**Table 5 diagnostics-11-01168-t005:** Patients’ characteristics of Study 2.

Characteristics	Overall	Severe OSA	Non-Severe OSA	Effect Size, Median of Difference (95% CI) or Odds Ratio (95% CI) ^1^	*p*-Value ^2^
	(*n* = 100)	(*n* = 47)	(*n* = 53)		
Demographic data					
Age (y)	7 (6–10)	8 (5–10)	7 (6–10)	0 (−1–1)	0.93
Sex (female/male)	27/73	15/32	12/41	1.6 (0.7–3.9)	0.37
BMI (kg/m^2^) z-score	0.6 (−0.5–1.9)	0.8 (−0.3–2.2)	0.2 (−0.6–1.8)	0.5 (−0.1–1.1)	0.07
Anatomical data					
Tonsil size (1‒4)	3 (3–4)	3 (3–4)	3 (3–4)	0 (0–0)	0.14
ANR	0.85 (0.66–0.91)	0.91 (0.72–0.94)	0.77 (0.60–0.90)	0.07 (0.03–0.14)	**<0.001**
Polysomnographic data					
Obstructive AHI (events/h)	9.6 (5.2–22.8)	24.4 (15.9–44.7)	5.3 (4.0–8.0)	18.4 (14.4–24.3)	**<0.001**
Obstructive AI (events/h)	2.3 (0.7–5.1)	5.2 (2.1–12.4)	1.3 (0.6–2.7)	3.7 (2.2–6.5)	**<0.001**

Data are summarized as median and interquartile range or number. Abbreviations: AHI: apnea–hypopnea index; AI: apnea index; ANR: adenoid-nasopharynx ratio; BMI: body mass index; OSA: obstructive sleep apnea. ^1^ Effect sizes were calculated with the use of Hodges–Lehmann method for Mann–Whitney *U* test, or odds ratio calculation for Fisher’s exact test. ^2^ Data were compared between the severe OSA subgroup and non-severe OSA subgroup using the Mann–Whitney *U* test or Fisher’s exact test as appropriate. Significant *p*-values are marked in bold.

**Table 6 diagnostics-11-01168-t006:** Predictive performance of the anatomical model and the combined model of severe OSA of Study 2.

	Receiver Operating Characteristic Curves				Predictive Performance					Cohen’s Kappa	
Predictive Models	Cut-Off Value	AUC	95% CI	*p*-Value	Sen	Spec	PPV	NPV	Acc	Kappa	*p*-Value
The anatomical model	≥1 risk factor	0.61	0.50–0.72	**0.049**	85%	38%	55%	74%	60%	0.22	**0.01**
The combined model	≥4	0.83	0.74–0.91	**<0.001**	77%	75%	73%	78%	76%	0.52	**<0.001**

Abbreviations: Acc: accuracy; AUC: area under the curve; CI: confidence interval; NPV: negative predictive value; PPV: positive predictive value; Sen: sensitivity; Spec: specificity; OR: odds ratio. Significant *p*-values are marked in bold.

## Data Availability

The data presented in this study are available on request from the corresponding author. The data are not publicly available due to ethical restrictions.

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
