# Peer review of "Screening Severe Obstructive Sleep Apnea in Children with Snoring"

_diagnostics, 2021, doi:10.3390/diagnostics11071168_

Round 1
Reviewer 1 Report
Hsieh HS et al report that snoring is a hallmark of pediatric OSA difficult to study with the actual screening tools. This retrospective study compared the efficiency of potential clinical predictors, home SSA, and home sleep PO in the screen of severe OSA in children with habitual snoring. The development of the best predictive model of severe pediatric OSA has also been approached. The Authors suggest that home sleep pulse oximetry, combined with ANR, can screen for severe OSA more efficaciously than ANR and tonsil size in children with snoring, and the Authors speculate “with reduced OSA-related health costs”.
As pointed out by the Authors in the limitations, this a retrospective study and a prospective one is needed to adequately investigate the issue. However, the work is interesting, clear and exhaustive.
Methodology: the work is well conducted, statistical analysis appropriated.
Points of revision: Clinical variables: it seems that direct inspection of the tonsils is a subjective decision (ref23). Could you explain how to minimize the possible differences between different operators?
Lines137-138: recording from 8 to 12PM to 6 PM. I don’t understand, could you check?
Results: clear and well explained. The difference obtained in a different age-population is a point of concern, which the Authors recognized in the limitations. A prospective study is needed to confirm these data in children of 5-12 years.
Discussion: well done.
Author Response
Reviewer 1’s comments:
Hsieh HS et al report that snoring is a hallmark of pediatric OSA difficult to study with the actual screening tools. This retrospective study compared the efficiency of potential clinical predictors, home SSA, and home sleep PO in the screen of severe OSA in children with habitual snoring. The development of the best predictive model of severe pediatric OSA has also been approached. The Authors suggest that home sleep pulse oximetry, combined with ANR, can screen for severe OSA more efficaciously than ANR and tonsil size in children with snoring, and the Authors speculate “with reduced OSA-related health costs”.
As pointed out by the Authors in the limitations, this a retrospective study and a prospective one is needed to adequately investigate the issue. However, the work is interesting, clear and exhaustive.
Methodology: the work is well conducted, statistical analysis appropriated.
REPLY. We appreciate these encouraging comments. We do our best and believe that this paper offers insight into the associations between ODI ≥3%, ANR, and severe OSA. The results of this study highlight promising future research pathways through which precision medicine protocols may be developed for pediatric patients with OSA. Thank you very much!
Points of revision: Clinical variables: it seems that direct inspection of the tonsils is a subjective decision (ref23). Could you explain how to minimize the possible differences between different operators?
REPLY. Thank you very much for this in-depth comment. Two senior otolaryngological investigators determined the tonsil size (Profs. Li-Ang Lee and Hsueh-Yu Li). We worked together for more than 15 years for the treatment of OSA. For minimizing the possible differences between operators, we have made a consensus on tonsil sizing since 2009 (Li, H.-Y., Lee, L.-A. Sleep-disordered breathing in children. Chang Gung Med J. 2009;32(3):247-57). We have evaluated intraobserver reliability several times before, and the intraclass correlation coefficients were approximate 0.87-0.93 using the Brodsky grading scale. Our intraobserver reliability was compatible with a previous report (mean interobserver reliability of the Brodsky grading scale: intraclass correlation coefficient = 0.721) (Kumar, D.S., Valenzuela, D., Kozak, F.K., Ludemann, J.P., Moxham, J.P., Lea, J., Chadha, N.K. The reliability of clinical tonsil size grading in children. JAMA Otolaryngol Head Neck Surg 2014;140(11):1034-7. doi: 10.1001/jamaoto.2014.2338.). We add this information in the revised manuscript.
Modified text, Page 3, lines 131-137
‘We directly inspected the tonsils and graded the tonsil size on a scale from 0–4 for each patient [23]. We also measured the adenoidal-nasopharyngeal ratio (ANR) to grade the adenoid size using lateral radiography of the nasopharynx [24].’
-->
‘We directly inspected the tonsils and graded the tonsil size on a scale from 0–4 for each patient using the Brodsky grading scale [26], which has been shown to have excellent interobserver and intraobserver reliability [27]. We also measured the adenoidal-nasopharyngeal ratio (ANR) to grade the adenoid size using lateral radiography of the nasopharynx [28]. For minimizing the possible differences between operators, two senior investigators (Lee, L.-A. and Li, H.-Y.) measured the tonsil size and ANR following the previous report [29].’
Lines137-138: recording from 8 to 12PM to 6 PM. I don’t understand, could you check?
REPLY. Thank you very much for pointing out this issue. In Taiwan, children frequently sleep at 8 to 12 PM and awake at 6 PM. Therefore, we can obtain oxygen saturation data during this period. For avoiding confusion, we amend it in the revised manuscript.
Modified text, Page 3, lines 150-151
‘… Seven of 10 subjects had ≥60% valid signal data for full-night recordings from 8 to 12 PM to 6 AM, whereas nine had ≥60% valid signal data for the first 150 min of recordings. …’
-->
‘… Seven of 10 subjects had ≥60% valid signal data for full-night recordings (range, 6–10 hours), whereas nine had ≥60% valid signal data for the first 150 min of recordings. …’
Results: clear and well explained. The difference obtained in a different age-population is a point of concern, which the Authors recognized in the limitations. A prospective study is needed to confirm these data in children of 5-12 years.
REPLY. Thank you very much for these in-depth comments. We add this important information in the revised limitation.
Modified text, Page 11, lines 422-427
‘The limitations of this study included (a) selection bias that might have existed concerning the single ethnicity, a predominance of boys, a subgroup of children with snoring, and a confined age spectrum of 5–12 years, and (b) the lack of a normal control group. However, the best predictive model reached acceptable accuracy in the subsequent validation cohort. Nevertheless, future studies are warranted for more details investigating the other variables of interest using a larger sample size to minimize selection bias.’
-->
‘… Second, selection bias that might have existed concerning the single ethnicity, a predominance of boys, a subgroup of children with snoring, a confined age spectrum of 5–12 years, and a lack of a normal control group. However, the best predictive model reached acceptable accuracy in the subsequent validation cohort. Nevertheless, further research with a prospective design and a larger sample size is warranted to confirm the findings of this study.’

Reviewer 2 Report
The main result of this study is the validation of a simple technique to predict severe sleep apnea in children. The sample size and methods is sufficient, results convicing.
Please accept in present form.
Author Response
REPLY. We appreciate these encouraging comments. We do our best and believe that this paper offers a simple technique to predict severe OSA in children. The results of this study highlight promising future research pathways through which precision medicine protocols may be developed for pediatric patients with OSA. Thank you very much!

Reviewer 3 Report
Overview:
The paper is a look at various screening methods for severe OSAS in children
There are a number of issues that need to be addressed:
General/Larger Issues:
- It is somewhat difficult to understand the ideas expressed in the paper due primarily to issues around grammatical and other language issues. – consider getting an English language editor.
- It is somewhat unclear how this adds much to the current literature. The findings are basically that SpO2 can be used to screen for OSAS, but SpO2 is part of the definition of sleep apnoea events and furthermore are very common in children – please explain how this isn’t just an autocorrelation. The fact that anatomical issues are poor correlates is already established as is the utility and limitations of snoring sound analysis.
- Introduction:
- it would be good to include a hypothesis or, if not, justify why.
- Neurocognitive and other impairments caused by OSAS is somewhat speculative or controversial in the literature and this should be acknowledged
- Methods:
- The PSG methodology as reported is not correct – define terms correctly (103-115) and if this was what was used then the stats will need to be rerun with the correct and preferably up to date rules
- Results:
- Report effect sizes
- Discussion:
- Report more fully the limitations
- Conclusion:
- none
Minor Language Issues:
- Examples of odd phrasing that make things difficult to understand are on:
- Line 36 – “ under the control of…” ??
- Line 52-53 – very odd sentence
- Line 83 – word missing?
- 90-91 – meaning unclear
- Line 327 – “the diversity could cause by..” ??
- others
Other Minor Issues
- The manuscript would also be more readable if the unnecessary acronyms were removed and their usage made consistent
- Refs missing in introduction
Author Response
Overview:
The paper is a look at various screening methods for severe OSAS in children
There are a number of issues that need to be addressed:
REPLY. Thank you very much for in-depth comments. We have made a point-by-point response to your comments and suggestions.
General/Larger Issues:
- It is somewhat difficult to understand the ideas expressed in the paper due primarily to issues around grammatical and other language issues. – consider getting an English language editor.
REPLY. Thanks for your cogent comments. A native English language author has edited this revised manuscript.
- It is somewhat unclear how this adds much to the current literature.
REPLY. Thank you very much for these in-depth comments. We confirmed that ODI3 had a better performance to predict severe OSA than the other variables of interest in children with loud snoring. Furthermore, the combination of ODI3 with the adenoidal-nasopharyngeal ratio provided the best accuracy to predict severe pediatric OSA. We hope our findings provide a practical perspective for clinicians to predict severe pediatric OSA more efficiently.
The findings are basically that SpO2 can be used to screen for OSAS, but SpO2 is part of the definition of sleep apnoea events and furthermore are very common in children – please explain how this isn’t just an autocorrelation.
REPLY. According to the 2012 American Academy of Sleep Medicine Scoring Manual, apneas are defined by ≥90% decrease in airflow for a duration of ≥2 breaths, and hypopneas are defined by ≥50% decrease in airflow and either ≥3% desaturation or electroencephalographic arousal for a duration of ≥2 breaths. Therefore, a change of SpO2 is not necessary of the definition of apneas but is part of the definition of hypopneas. Accordingly, the oxygen desaturation index (ODI3) is not equal to the apnea-hypopnea index (AHI). Furthermore, nocturnal pulse oximetry detected oxygen desaturation events without considering the respiratory events. In this study, the ODI3 was detected by nocturnal pulse oximetry in a home environment, whereas the AHI was detected by standard polysomnography in a sleep laboratory. Although ODI3 and AHI were partially autocorrelated at the same condition in standard polysomnography, the significant association of ODI3 and AHI in this study was not an autocorrelation due to different definitions, detectors, and sleep environments. We have amended this information in the study limitation.
Modified text, Page 11, lines 416-423
‘The limitations of this study included (a) selection bias that might have existed concerning the single ethnicity, …’
-->
‘Our study had some limitations. First, oxygen desaturation was part of the definition of hypopneas; therefore, the ODI3 might be partially autocorrelated with the AHI. However, most nocturnal pulse oximetry detected oxygen desaturation events without considering the respiratory events. Furthermore, we obtained the ODI3 using nocturnal pulse oximetry in a home environment and the AHI using standard polysomnography in a sleep laboratory. Therefore, the significant association of ODI3 and AHI in this study was not an autocorrelation due to different definitions, detectors, and sleep environments. Second, selection bias that might have existed concerning the single ethnicity, …’
The fact that anatomical issues are poor correlates is already established as is the utility and limitations of snoring sound analysis.
REPLY. Thank you for this cogent comment. In this study, we confirmed that anatomical issues and snoring sound analysis were not effective screening tool for severe OSA in children with snoring.
Modified text, Page 2, lines 70-73
‘… Even though the pathophysiologies behind these rapid screening tests are distinctly different, they can all be used to predict the severity of OSA. However, to the best of our knowledge, no previous study has …’
-->
‘… Even though the pathophysiology behind these rapid screening tests is distinctly different, they can all be used to predict the severity of OSA. The utility and limitations of anatomy-based classification systems and snoring sound analysis have been elucidated in some previous research. Nevertheless, to the best of our knowledge, no previous study has …’
- Introduction:
- it would be good to include a hypothesis or, if not, justify why.
REPLY. Thank you for this excellent comment. We have added a hypothesis in the introduction.
Modified text, Page 2, lines 78-84
‘The first aim of this retrospective study was to compare the efficiency of potential clinical predictors, home SSA, and home sleep PO in the screen of severe OSA in children with habitual snoring, and secondly to develop the best predictive model of severe pediatric OSA by using these convenient screening approaches to undergo further prompt allocating patients with timely managements.’
-->
‘We hypothesized that clinical predictors, home snoring sound analysis, and home sleep pulse oximetry had different effectiveness in screening severe OSA among children with habitual snoring. The first aim of the study was to compare the differences in the effectiveness of clinical predictors, home snoring sound analysis, and home sleep pulse oximetry in screening severe pediatric OSA. The second aim of the study was to develop the best predictive model for severe pediatric OSA by using these highly available screening approaches to facilitate timely diagnosis and early intervention.’
- Neurocognitive and other impairments caused by OSAS is somewhat speculative or controversial in the literature and this should be acknowledged.
REPLY. Thank you for this in-depth comment. We have added this information in the introduction.
Modified text, Page 2, lines 48-50
‘… Pediatric OSA is a possible cause of neurocognitive/behavioral impairment and cardiovascular/metabolic dysfunction and ultimately compromises overall health and quality of life [1,2]. Adenotonsillectomy …’
-->
‘… A substantial volume of evidence has linked pediatric OSA and cardiovascular/metabolic dysfunctions [1,2]. Some research has also suggested a connection between pediatric OSA and neurocognitive/behavioral impairment, but the results are still inconsistent and inconclusive [3]. Adenotonsillectomy …’
- Methods:
- The PSG methodology as reported is not correct – define terms correctly (103-115) and if this was what was used then the stats will need to be rerun with the correct and preferably up to date rules.
REPLY. Thank you for these cogent comments. We are sorry for these typos and have amended the AHI, obstructive AHI, and AHI definitions in the revised manuscript.
Modified text, Page 3, lines 117-125
‘The apnea-hypopnea index (AHI) was defined as the sum of all obstructive and mixed apneas (≥90% decrease in airflow for a duration of ≥2 breaths), plus hypopneas (≥50% decrease in airflow and either ≥3% desaturation or electroencephalographic arousal, for a duration of ≥2 breaths), divided by the number of hours of total sleep time, according to the 2012 American Academy of Sleep Medicine Scoring Manual [20]. The AHI and apnea index (AI) were recorded for further comparisons. In the present study, the patients were categorized as having ‘severe’ (obstructive AHI ≥ 10.0 events/h) or ‘non-severe’ (obstructive AHI < 10.0 events/h) OSA [21].’
-->
‘The apnea-hypopnea index (AHI) was defined as the sum of all apneas (≥90% decrease in airflow for a duration of ≥2 breaths) plus hypopneas (≥50% decrease in airflow and either ≥3% desaturation or electroencephalographic arousal, for a duration of ≥2 breaths), divided by the number of hours of total sleep time, according to the 2012 American Academy of Sleep Medicine Scoring Manual [21]. The obstructive AHI and apnea index (AI) were recorded for further comparisons. In the present study, the diagnosis of OSA was defined by an OAHI ≥ 2.0 events/h or an obstructive apnea index (OAI) ≥ 1.0 events/h [22,23]. Furthermore, the patients were categorized as having ‘severe’ (obstructive AHI ≥10.0 events/h) or ‘non-severe’ (obstructive AHI <10.0 events/h) OSA [24].’
- Results:
- Report effect sizes
REPLY. Thank you for this cogent comment. We have reported effect sizes in the modified Tables 1–3.
Modified text, Page 5, lines 224-228
‘… The severe OSA sub-group had a significantly larger tonsil size, higher ANR, and higher obstructive AHI compared to the non-severe OSA subgroup. There were no significant differences in age, sex, BMI z-score, OSA-18, and obstructive AI.’
-->
‘… The severe OSA sub-group had a significantly larger tonsil size (effect size = 1; 95% confidence interval [CI]: 0–1), higher ANR (effect size = 0.12; 95% CI: 0.06–0.19), and higher obstructive AHI (effect size = 19.0; 95% CI: 13.8–27.8) compared to the non-severe OSA subgroup. There were no significant differences in age, sex, BMI z-score, OSA-18, and obstructive AI.’
Modified text, Page 6, lines 241-247
‘… The severe OSA subgroup had a significantly higher ANR (effect size = 0.07; 95% CI: 0.03–0.14), higher obstructive AHI, and higher obstructive AI compared to the non-severe OSA subgroup. Furthermore, the median age of the validation cohort was significantly lower than that of the discovery cohort, whereas the distributions of sex, BMI z-score, tonsil size, ANR, AHI, and AI of both cohorts were compatible (all p > 0.05).’
-->
‘… The severe OSA subgroup had a significantly higher ANR (effect size = 0.07; 95% CI: 0.03–0.14), higher obstructive AHI (effect size = 18.4; 95% CI: 14.4–24.3), and higher obstructive AI (effect size = 3.7; 95% CI: 2.2–6.5) compared to the non-severe OSA subgroup. Furthermore, the median age of the validation cohort was significantly lower than that of the discovery cohort (effect size = -1; 95% CI: -2–0; p = 0.02), whereas the distributions of sex, BMI z-score, tonsil size, ANR, AHI, and AI of both cohorts were compatible (all p > 0.05).’
Modified text, Page 7, lines 263-272
‘… The severe OSA subgroup had significantly a higher ODI3 compared to the non-severe OSA subgroup in both cohorts.
Using the 6-h SSA test, all patients had available data ≥90% to analyze the measurements taken at home. In the discovery cohort, the severe OSA subgroup had a significantly higher SSE of 801–1000 Hz than the non-severe subgroup. The differences in other SSA variables between the subgroups were not significant.’
-->
‘… The severe OSA subgroup had significantly a higher ODI3 compared to the non-severe OSA subgroup in both cohorts (effect size = 7.3; 95% CI: 3.5–20.1).
Using the 6-h SSA test, all patients had available data ≥90% to analyze the measurements taken at home. In the discovery cohort, the severe OSA subgroup had a significantly higher SSE of 801–1000 Hz than the non-severe subgroup (effect size = 9; 95% CI: 0–20.3). The differences in other SSA variables between the subgroups were not significant.
In the validation cohort, the severe OSA subgroup had significantly a higher ODI3 compared to the non-severe OSA subgroup in both cohorts (effect size = 16.9; 95% CI: 12.3–16.9).’
- Discussion:
- Report more fully the limitations
REPLY. Thanks for this in-depth comment. We have reported more fully the study limitations.
Modified text, Page 11, lines 416-427
‘The limitations of this study included (a) selection bias that might have existed concerning the single ethnicity, a predominance of boys, a subgroup of children with snoring, and a confined age spectrum of 5–12 years, and (b) the lack of a normal control group. However, the best predictive model reached acceptable accuracy in the subsequent validation cohort. Nevertheless, future studies are warranted for more details investigating the other variables of interest using a larger sample size to minimize selection bias.’
-->
‘Our study had some limitations. First, oxygen desaturation was part of the definition of hypopneas; therefore, the ODI3 might be partially autocorrelated with the AHI. However, most nocturnal pulse oximetry detected oxygen desaturation events without considering the respiratory events. Furthermore, we obtained the ODI3 using nocturnal pulse oximetry in a home environment and the AHI using standard polysomnography in a sleep laboratory. Therefore, the significant association of ODI3 and AHI in this study was not an autocorrelation due to different definitions, detectors, and sleep environments. Second, selection bias that might have existed concerning the single ethnicity, a predominance of boys, a subgroup of children with snoring, a confined age spectrum of 5–12 years, and a lack of a normal control group. However, the best predictive model reached acceptable accuracy in the subsequent validation cohort. Nevertheless, further research with a prospective design and a larger sample size is warranted to confirm the findings of this study.
- Conclusion:
- none
REPLY. Thank you very much for this excellent comment.
Minor Language Issues:
- Examples of odd phrasing that make things difficult to understand are on:
- Line 36 – “ under the control of…” ??
REPLY. Amended.
Modified text, Page 1, lines 35-36
‘… Multivariate analysis showed that ODI3 ≥6.0 events/h independently predicted severe pediatric OSA under the control of ANR ≥0.78. …’
-->
‘… Multivariate analysis showed that ODI3 ≥6.0 events/h independently predicted severe pediatric OSA. …’
- Line 52-53 – very odd sentence
REPLY. Amended.
Modified text, Page 1, lines 53-55
‘… Consequently, the delayed diagnosis of severe pediatric OSA is of the utmost importance to sub-optimize treatment in children with sleep-disordered breathing.’
-->
‘… Therefore, a delay in the diagnosis of severe pediatric OSA has been one of the major attributable causes for compromised treatment outcomes for pediatric sleep-disordered breathing.’
- Line 83 – word missing?
REPLY. Amended.
Modified text, Page 2, lines 87-90
‘This retrospective cross-sectional study was approved by the Institutional Review Board of the Chang Gung Memorial Foundation (No. 202000873B0). All procedures were carried following the current regulations. The requirement for written informed consent was waived.’
-->
‘This retrospective cross-sectional study was approved by the Institutional Review Board of the Chang Gung Memorial Foundation (No. 202000873B0). The study was conducted according to the guidelines of the Declaration of Helsinki. The requirement for written informed consent was waived.’
- 90-91 – meaning unclear
REPLY. Amended.
Modified text, Page 2, lines 94-98
‘… The first 42 consecutive subjects with habitual snoring (snoring three or more times per week) underwent nocturnal PO and SSA at home for system trained as a discovery cohort, while the remaining 108 consecutive subjects with snoring were enrolled for nocturnal PO as a validation cohort. …’
-->
‘… The first 42 consecutive subjects with habitual snoring (snoring three or more times per week) underwent nocturnal pulse oximetry and snoring sound analysis at home for system trained as a discovery cohort. In comparison, the remaining 108 consecutive subjects with snoring were enrolled for nocturnal pulse oximetry as a validation cohort. …’
- Line 327 – “the diversity could cause by..” ??
REPLY. Amended.
Modified text, Page 10, line 359
‘… The diversity could cause by varied sample size and screening of the …’
-->
‘… The diversity could be caused by varied sample sizes and screening of the …’
- others
REPLY. Thank you very much for this excellent comment.
Other Minor Issues
- The manuscript would also be more readable if the unnecessary acronyms were removed and their usage made consistent.
REPLY. Thanks for this cogent comment. We reduced the use of acronyms such as PSG, PO, SSA, and SI. We found the revised manuscript was more readable. Thank you very much!
- Refs missing in introduction
REPLY. Amended.
Modified text, Page 2, lines 48-49
‘… approximately affects 2 to 3% of school-aged children. Pediatric OSA is a possible cause of neurocognitive/behavioral impairment and cardiovascular/metabolic dysfunction and ultimately compromises overall health and quality of life [1,2]. …’
-->
‘… approximately affects 2 to 3% of school-aged children [1,2]. A substantial volume of evidence has linked pediatric OSA and cardiovascular/metabolic dysfunctions [1,2]. Some …’

Reviewer 4 Report
Screening severe obstructive sleep apnea in children with snoring by Hui-Shan Hsieh et al. approaches a hot topic area in pediatric sleep medicine. It targets to develop a novel approach of a combination tests for a simpler method of screening. But there are some issues that can be solved.
- page 3, line 111. Standard- in laboratory full-night polysomnography. Is it one night only?
- page 3, line 124. Did you validate PSG manually for AHI? The same certified technician?
- page 9, line 320. Can you imagine a computer based prediction score that combines the best predictors from your paper?
- Main limitation: sample size, retrospective cross-sectional.
Author Response
Reviewer 4:
Screening severe obstructive sleep apnea in children with snoring by Hui Shan Hsieh et al. approaches a hot topic area in pediatric sleep medicine. It targets to develop a novel approach of a combination tests for a simpler method of screening. But there are some issues that can be solved.
REPLY. We appreciate your in-depth and encouraging comments. We do our best and believe that this paper offers a novel approach of a combination testes for a simpler method of screening severe OSA in children with loud snoring. We have made a point-by-point response to your comments and suggestions and marked the changes made from the previous article file as a revised manuscript file. Your comments and suggestions have substantially improved the quality of the study. Thank you very much!
- page 3, line 111. Standard- in laboratory full-night polysomnography. Is it one night only?
REPLY.
Yes, the participants underwent one standard in-laboratory full-night polysomnography. Because this study was aimed to investigate the nocturnal to reduce respiratory pattern (i.e., OSA severity), we did not arrange an adaption night for these pediatric patients according to a previous study (doi: 10.1016/s1388-2457(03)00209-8). However, there was a first night effect in children with suspected OSA undergoing polysomnography, an adaption night was necessary when sleep architecture was to be studied. We have added the information in the study limitation. Thank you very much!
Modified text, Pages 12-13, lines 468-472
‘… different definitions, detectors, and sleep environments. Second, selection bias …’
-->
‘… of the definition of hypopneas. Third, an adaption night standard polysomnography was not included in our protocol. Although an adaption night is considered necessary only for nocturnal respiratory pattern investigations, the first-night effect may still influence the sleep architectures in subjects [69]. Forth, selection bias …’
- page 3, line 124. Did you validate PSG manually for AHI? The same certified technician?
REPLY.
Yes, the study investigators (Huang, Y.-S., Chuang, L.-P.) validated polysomnography manually for AHI. The same registered sleep technologists performed these polysomnograms to ensure the quality of interpretation. We have added the information in the revised manuscript.
Modified text, Page 3, lines 128-131
‘… The oximeters employed a sampling frequency of 0.5 Hz.
The apnea-hypopnea index (AHI) was defined as the sum of all apneas …’
-->
‘… The oximeters employed a sampling frequency of 0.5 Hz. The same registered sleep technologists performed polysomnograms, and the study investigators (Huang, Y.-S. and Chuang, L.-P.) validated polysomnography manually to ensure the quality of interpretation.
The apnea-hypopnea index (AHI) was defined as the sum of all apneas …’
- page 9, line 320. Can you imagine a computer-based prediction score that combines the best predictors from your paper?
REPLY.
We appreciate you for bringing this important issue to us. In the revised manuscript, we further used multivariate categorical regression analyses with the LASSO (doi:10.1111/j.2517-6161.1996.tb02080.x) to optimal select variables and improve predictive performance for predicting severe pediatric OSA of Study 1. We found the combined model included ODI3 ≥6.0 events/h, tonsil size = 4, and ANR ≥0.78. For weighting the regression coefficient, the total score was calculated by (the presence of ODI3 ≥6.0 events/h) × 3 + (the presence of tonsil size = 4) × 2 + (the presence of ANR ≥0.78) × 1. Furthermore, the combined model achieved an AUC of 0.93 (≥4 scores; p <0.001) with a sensitivity of 91% and a specificity of 94%. The PPV was 95%, and the NPV was 89% in this model the highest accuracy of 92%. By utilizing the combined model of Study 2, the accuracy of prediction was 76% and the Kappa was 0.52 (p <0.001). We have added the information in the revised Abstract, Materials and Methods, Results, and Conclusions. Thanks for your excellent suggestions.
Modified text, Page 1, lines 36-41
‘… Among several predictive models, the combination of ODI3 and ANR best screened for severe OSA with a sensitivity of 81% and a specificity of 94%. This model predicted severe OSA with an accuracy of 72% in the validation cohort (27 [27%] girls and 73 [73%] boys; median age, 7 years). Our results suggested that home sleep pulse oximetry, combined with ANR, can screen for severe OSA more effectively than ANR and tonsil size among children with snoring.’
-->
‘… Among several predictive models, the combination of ODI3, tonsil size, and ANR more optimally screened for severe OSA with a sensitivity of 91% and a specificity of 94%. Of Study 2 (27 [27%] girls and 73 [73%] boys; median age, 7 years), this model was externally validated to predict severe OSA with an accuracy of 76%. Our results suggested that home sleep pulse oximetry, combined with ANR, can screen for severe OSA more optimally than ANR and tonsil size among children with snoring.’
Modified text, Page 5, lines 215-219
‘… The dichotomized variables were then assessed using univariate and multivariate logistic regression models. All p-values were two-sided, and statistical significance was accepted at p <0.05. …’
-->
‘… The dichotomized variables were then assessed using univariate and multivariate logistic regression models. For selecting optimal variables and improving predictive performance, multivariate categorical regression models of all dichotomized variables with the logistic least absolute shrinkage and selection operator (LASSO) and bootstrap resampling (n of samples = 50; 100 runs) were performed [41]. All p-values were two-sided, and statistical significance was accepted at p <0.05. …’
Modified text, Page 8, lines 301-304
‘Using multivariate logistic regression analysis, we constructed several prediction models, including an anatomical model, a home sleep apnea test model, and the best predictive model (Table 5).’
-->
‘Using multivariate logistic regression analysis, we constructed several prediction models, including an anatomical model and a home sleep apnea test model (Table 4). Using multivariate categorical regression analysis with the LASSO (Figure 2), the combined predictive model was constructed.’
Modified text, Pages 8-9, Table 4
Add
-->
|
The combined predictive model |
Total score = (ODI3 ≥6.0 events/h) × 3 + (Tonsil size = 4) × 2 + (ANR ≥0.78) × 1 |
|||||||||||
|
|
Beta |
SE |
p-value |
Cut-off value |
AUC |
95% CI |
p-Value |
Sen |
Spec |
PPV |
NPV |
Acc |
|
ODI3 ≥6.0 events/h |
0.61 |
0.15 |
<0.001 |
≥4 |
0.93 |
0.83–1.00 |
<0.001 |
91% |
94% |
95% |
89% |
92% |
|
Tonsil size = 4 |
0.21 |
0.13 |
0.03 |
|
|
|
|
|
|
|
|
|
|
ANR ≥0.78 |
0.27 |
0.11 |
0.13 |
|
|
|
|
|
|
|
|
|
Modified text, Page 9, Figure 2
Add
-->
Figure 2. Logistic least absolute shrinkage and selection operator paths for severe pediatric obstructive sleep apnea (OSA). Five variables, including oxygen desaturation index ≥3% (ODI3) ≥6.0 events/h, adenoid/nasopharynx ratio (ANR) ≥0.78, tonsil size -4, snoring sound energy (SSE) of 801–1000 Hz ≥22.0 dB, and OSA-18 ≥77, were analyzed using the multivariate categorical regression models. The X-axis reference line indicates the combined model (three variables).
Modified text, Page 9, lines 327-332
‘The best predictive model included ODI3 ≥6.0 events/h and ANR ≥0.78 as predictors for severe OSA. Therefore, the best predictive model achieved an AUC of 0.88 (≥1 predictor; p <0.001) with a sensitivity of 81% and a specificity of 94% (≥1 predictor). The PPV was 94%, and the NPV was 81% in this model the highest accuracy of 87%.’
-->
‘The combined model included ODI3 ≥6.0 events/h, tonsil size = 4, and ANR ≥0.78 as predictors for severe OSA. For weighting the regression coefficient, (the presence of ODI3 ≥6.0 events/h) × 3 + (the presence of tonsil size = 4) × 2 + (the presence of ANR ≥0.78) × 1. Therefore, the combined model achieved an AUC of 0.93 (≥4 scores; p <0.001) with a sensitivity of 91% and a specificity of 94%. The PPV was 95%, and the NPV was 89% in this model the highest accuracy of 92%.’
Modified text, Page 10, lines 368-370
‘By utilizing the anatomical model in the validation cohort, the accuracy of prediction was 61% and the Kappa was 0.22 (p = 0.01) (Table 6). By utilizing the best predictive model, the accuracy of prediction was 72% and the Kappa was 0.46 (p <0.001).’
-->
‘The prediction accuracy of the anatomical model was 61% and the Kappa was 0.22 (p = 0.01) (Table 6). The prediction accuracy of the combined model was 76% and the Kappa was 0.56 (p <0.001).’
Modified text, Page 13, lines 483-485
‘Furthermore, the combination of ODI3 with the ANR provided the best accuracy to predict severe pediatric OSA.’
-->
‘Furthermore, the combination of ODI3 with the tonsil size and ANR might perform better in predicting severe pediatric OSA.’
- Main limitation: sample size, retrospective cross-sectional.
REPLY.
We appreciate your cogent comment. We acknowledge that the main limitation of the retrospective cross-sectional study design with the given small sample size. We have added this information in the revise manuscript.
Modified text, Pages 12-13, lines 464-477
‘Our study had some limitations. First, oxygen desaturation was part of the definition of hypopneas; therefore, the ODI3 might be partially autocorrelated with the AHI. … Second, selection bias that might have existed concerning the single ethnicity, a predominance of boys, … However, the best predictive model reached acceptable accuracy in the subsequent validation cohort. Nevertheless, further research with a prospective design and a larger sample size is warranted to confirm the findings of this study.’
-->
‘Our study had some limitations. First, the study was retrospective cross-sectional and with a small sample size. … Nevertheless, the combined model reached a considerable accuracy in the subsequent Study 2 cohort. Further prospective investigation with a sample larger in size and more heterogenous in demographics (e.g., different ethnic groups) is warranted to verify the findings of this research.’

Reviewer 5 Report
This is a retrospective cross-sectional study aimed to compare clinical variables, home snoring sound analysis, and home sleep pulse oximetry on their predictive performance in screening for severe OSA among children who snored.
Considering 150 patients who underwent standard full night polysomnography (PSG), 39 (out 42 participants) were also recorded at home for some clinical variables and unattended nocturnal SpO2 and snoring signals.
The remaining 100 (out 108 partipants) were recorded for nocturnal pulse oximetry.
The results suggested that home sleep pulse oximetry, combined with nasopharyngeal ratio can screen for severe OSA more effectively than nasopharyngeal ratio and tonsil size among children with snoring.
Four points should be mandatorily addressed in the revision:
- The first and major issue is that the design of the study is not competely clear. If I understand, starting from the standard PSG for all 150 patients, 42 were recorded at home and 108 were recorded for nocturnal pulse oximetry. What is unclear concerns the second group (the so-called "validation group"). Was this group recorded again in standard in laboratory? Or, alternatively, were their data of the original standard PSG considered?
- Why within group analyses were not carried out? In other words, the 39 patients of the so-called "discovery group" also had standard PSG. This means that they had measures of standard nocturnal SpO2 and snoring activity both in laboratory and at home. This comparison is a more stingent test of the validity and reliability of home monitoring.
- Across the whole manuscript, the clinical routine of cardiopulmonary measures collected at home seems completely neglected. These sleep studies have been shown to be an accurate and practical alternative to overnight laboratory polysomnography (Kirk V, Kahn A, Brouillette RT. Diagnostic approach to obstructive sleep apnea in children. Sleep Med Rev. 1998 Nov;2(4):255-69. doi: 10.1016/s1087-0792(98)90012-0)
- Sample size of the "discovery group" is relatively small
Author Response
Reviewer 5:
This is a retrospective cross-sectional study aimed to compare clinical variables, home snoring sound analysis, and home sleep pulse oximetry on their predictive performance in screening for severe OSA among children who snored. Considering 150 patients who underwent standard full night polysomnography (PSG), 39 (out 42 participants) were also recorded at home for some clinical variables and unattended nocturnal SpO2 and snoring signals. The remaining 100 (out 108 partipants) were recorded for nocturnal pulse oximetry. The results suggested that home sleep pulse oximetry, combined with nasopharyngeal ratio can screen for severe OSA more effectively than nasopharyngeal ratio and tonsil size among children with snoring.
REPLY. We appreciate your in-depth and encouraging comments. We do our best and believe that this paper offers a simple approach to predict severe OSA in children with loud snoring. We have made a point-by-point response to your comments and suggestions and marked the changes made from the previous article file as a revised manuscript file. Your comments and suggestions have substantially improved the quality of the study. Thank you very much!
Four points should be mandatorily addressed in the revision:
- The first and major issue is that the design of the study is not competely clear. If I understand, starting from the standard PSG for all 150 patients, 42 were recorded at home and 108 were recorded for nocturnal pulse oximetry. What is unclear concerns the second group (the so-called "validation group"). Was this group recorded again in standard in laboratory? Or, alternatively, were their data of the original standard PSG considered?
REPLY.
We appreciate these cogent comments. Sorry for the confusion and thank you for giving us an opportunity to clarify.
The research comprised two parts: Study 1 and 2.
The Study 1 (discovery study) was conducted to achieve the two aims of this research, namely:
- to compare the predictive performance of clinical variables, home snoring sound analysis, and home sleep pulse oximetry in screening severe pediatric OSA, and
- to develop a combined predictive model for severe pediatric OSA with measures from these existing screening tools to facilitate timely diagnosis and early intervention.
To ensure the internal validity of the research, a case-matched control design was adopted to reduce the confounding effects from variables known to be associated with OSA severity.
The Study 2 (validation study) was conducted to exam the external validity of the research. Predictive performance of the models was examined for sensitivity, specificity, positive predictive value, negative predictive value, and prediction accuracy.
Patients of Study 1 received “in-laboratory” polysomnography, from which the diagnosis and severity of OSA was confirmed, and then they received “home” sleep apnea tests, including nocturnal pulse oximetry and snoring sound analysis at home, and routine clinical evaluations. Predictive performance of various tools was examined, and three predictive models were developed.
In Study 2, the pediatric patients with loud snoring have received nocturnal pulse oximetry at home to assess severe OSA risk and then undergone standard in-laboratory polysomnography to confirm whether they have severe OSA or not.
We’ve revised the Materials and Methods section to better present the design of our research. Thank you again for helping us improve our work!
Modified text, Pages 2-3, lines 94-113
‘… Branch (Taoyuan, Taiwan) between March 1, 2015, and Jun 31, 2019. The first 42 consecutive subjects with habitual snoring (snoring three or more times per week) underwent nocturnal pulse oximetry and snoring sound analysis at home for system trained as a discovery cohort. In comparison, the remaining 108 consecutive subjects with snoring were enrolled for nocturnal pulse oximetry as a validation cohort. In the discovery cohort, there were 23 children with severe OSA and 19 age-, sex-, and BMI-matched control children with non-severe OSA.’
-->
‘… Branch (Taoyuan, Taiwan) between March 1, 2015, and Jun 31, 2019.
The investigation comprised two parts: Study 1 (discovery study) and Study 2 (validation study).
2.2.1. Study 1: Development of a Combined Model for Screening Severe Pediatric OSA
Data obtained from the first 42 consecutive subjects with habitual snoring (snoring three or more times per week) were used to compare the predictive performance of clinical variables, home snoring sound analysis, and home sleep pulse oximetry in screening severe pediatric OSA. A combined predictive model was further developed with measures from the abovementioned existing screening tools.
All the 42 patients had a standard in-laboratory polysomnography before home sleep apnea tests [9]. A case-matched control design was adopted to reduce the confounding ef-fects from variables known to be associated with OSA severity [20]. There were 23 children with severe OSA and 19 age-, sex-, and BMI-matched control children with non-severe OSA.
2.2.2. Study 2: External Validation of the Combined Model for Screening Severe Pediatric OSA
The remaining 108 consecutive subjects with snoring were included in the external validation of the combined model. Each of them underwent nocturnal pulse oximetry at home and subsequently received one standard in-laboratory polysomnography to confirm the diagnosis and severity of OSA.’
- Why within group analyses were not carried out? In other words, the 39 patients of the so-called "discovery group" also had standard PSG. This means that they had measures of standard nocturnal SpO2 and snoring activity both in laboratory and at home. This comparison is a more stingent test of the validity and reliability of home monitoring.
REPLY.
We appreciate your in-depth comments and suggestions. We have added within-group analyses of Study 1 in the revised Methods and Results.
Both home and in-laboratory pulse oximeters used the technology of plethysmograph. The home-ODI3 was significantly related to the polysomnography-ODI3 (r = 0.80, p < 0.001).
The snoring sound analysis detected snoring sounds whereas polysomnography detected snoring vibrations. Therefore, the home-snoring index (SI) was not statistically significantly associated with the polysomnography-SI (r = 0.24, p = 0.15) due to the differently given nature of measurements.
Thank you very much!
Modified text, Page 3, lines 136-138
‘… The obstructive AHI and apnea index (AI) were recorded for further comparisons. …’
-->
‘… The obstructive AHI, apnea index (AI), the number of SpO2 drops ≥3% per hour of recording (ODI3), and snoring index (SI) were recorded for further comparisons. …’
Modified text, Page 6, lines 241-246
‘… The severe OSA subgroup had a significantly larger tonsil size (effect size = 1; 95% confidence interval (CI): 0–1), higher ANR (effect size = 0.12; 95% CI: 0.06–0.19), and higher obstructive AHI (effect size = 19.0; 95% CI: 13.8–27.8), compared to the non-severe OSA subgroup. …’
-->
‘… The severe OSA subgroup had a significantly larger tonsil size (effect size = 1; 95% confidence interval (CI): 0–1), higher ANR (effect size = 0.12; 95% CI: 0.06–0.19), higher obstructive AHI (effect size = 19.0; 95% CI: 13.8–27.8), ODI3 (effect size = 18.9; 95% CI: 14.1–25.7), and SI (effect size = 160.0; 95% CI: 56.0–295.0), compared to the non-severe OSA subgroup. …’
Modified text, Page 7, lines 261-265
‘… severe and non-severe OSA subgroups.’
-->
‘… severe and non-severe OSA subgroups. The home-ODI3 was significantly related to the polysomnography-ODI3 (r = 0.80, p < 0.001) using the same plethysmograph technology. However, home-SI was not statistically significantly associated with the polysomnography-SI (r = 0.24, p = 0.15) with the difference in the given nature of measurements (snoring sound analysis: SSE; polysomnography: snoring vibration energy).’
- Across the whole manuscript, the clinical routine of cardiopulmonary measures collected at home seems completely neglected. These sleep studies have been shown to be an accurate and practical alternative to overnight laboratory polysomnography (Kirk V, Kahn A, Brouillette RT. Diagnostic approach to obstructive sleep apnea in children. Sleep Med Rev. 1998 Nov;2(4):255-69. doi: 10.1016/s1087-0792(98)90012-0)
REPLY.
Thank you for the important comments. According to the suggested systematic review (doi: 10.1016/s1087-0792(98)90012-0), Kirk et al. recommended that “home cardiopulmonary sleep studies were shown to be an accurate and practical alternative to overnight laboratory polysomnography for routine evaluation of non-complex children with adenotonsillar hypertrophy in Montreal (doi: 10.1002/ppul.1950200407). In the cited reference (doi: 10.1002/ppul.1950200407), Jacob et al. had used a simplified cardiorespiratory montage plus video recording to evaluate OSAS in pediatric patients with adenotonsillar hypertrophy. Recently, the high performances of level 2 (doi: 10.5664/jcsm.3970) and level 3 (doi: 10.5664/jcsm.7764) portable monitors to diagnosis pediatric OSA in school-aged children have been proved. However, Bhattacharjee commented that a failure rate (~18%) was relatively high for studies that were attended by night technologists and for older children (10.5664/jcsm.7748). Most study failures were related to poor oximetry signal and oxygen desaturation artifact (doi: 10.1378/chest.123.1.96; doi: 10.1542/peds.2017-3382; doi: 10.5664/jcsm.7764). Although a recent study placed emphasis on accurate oximetry signals (doi:10.1164/rccm.201705-0930OC), we found that the combination of home-ODI3 and ANR was more optimal than the solitary home-ODI3. We have added the information in the modified Discussion.
Modified text, Pages 11, lines 377-398
‘4. Discussion
Home sleep apnea tests with technically adequate devices have been used to diagnose moderate to severe OSA in uncomplicated adult patients with sleep-disturbed breathing [38]; … A familiar environment and simple devices may minimize the interference on one’s usual sleep and also improve children’s compliance. In this study, the ODI3 might provide greater accuracy than the SSE of 801–1000 Hz did.
Home sleep pulse oximetry is a promising tool to screen for OSA … However, easy dislodgement of the finger sensor is a major disadvantage of unattended pulse oximetry. Approximate 10% to 20% of hospital sleep pulse oximetry failed to provide technically acceptable data due to poor contact between the sensor and the child’s finger [32,46].’
-->
‘4. Discussion
Home sleep apnea tests with technically adequate devices have been used to diagnose moderate to severe OSA in uncomplicated adult patients [42] and school-aged children [43,44] with sleep-disturbed breathing; … A familiar environment and simple devices may minimize the interference on one’s usual sleep and also improve children’s compliance [47]. A combination of simplified cardiorespiratory montage and video recording have been proven to accurately evaluate OSA in pediatric patients with adenotonsillar hypertrophy [48]. In this study, the ODI3 might provide greater accuracy than the SSE of 801–1000 Hz did.
Home sleep pulse oximetry is a promising tool to screen for OSA … However, easy dislodgement of the finger sensor is a major disadvantage of unattended pulse oximetry. Approximate 10% to 20% of hospital and home sleep pulse oximetry failed to provide technically acceptable data due to poor contact between the sensor and the child’s finger [35,44,54]. Therefore, Bhattacharjee commented that the failure rate of home pulse oximetry and cardiorespiratory measures was relatively high for studies that were attended by night technologists and for older children [55].’
- Sample size of the "discovery group" is relatively small
REPLY.
Thank you for this cogent comment. We’ve revised the last paragraph in Discussion to acknowledge these study limitations from a retrospective cross-sectional study design and small sample size.
Modified text, Pages 12-13, lines 464-477
‘Our study had some limitations. First, oxygen desaturation was part of the definition of hypopneas; therefore, the ODI3 might be partially autocorrelated with the AHI. … Second, selection bias that might have existed concerning the single ethnicity, a predominance of boys, … However, the best predictive model reached acceptable accuracy in the subsequent validation cohort. Nevertheless, further research with a prospective design and a larger sample size is warranted to confirm the findings of this study.’
-->
‘Our study had some limitations. First, the study was retrospective cross-sectional and with a small sample size. … Nevertheless, the combined model reached a considerable accuracy in the subsequent Study 2 cohort. Further prospective investigation with a sample larger in size and more heterogenous in demographics (e.g., different ethnic groups) is warranted to verify the findings of this research.’

Round 2
Reviewer 3 Report
all comments appropriately addressed
Author Response
Dear Reviewer 3,
We appreciate your in-depth and encouraging comments. We do our best and believe that this paper offers a simple approach to predict severe OSA in children with loud snoring. Your comments and suggestions have substantially improved the quality of the study. Thank you very much!
Reviewer 5 Report
The authors made a serious effort to respond to the points I raised. In my opinion, it is now acceptable for publication
Author Response
We appreciate your encouraging comments, again. We do our best and believe that this paper offers a simple approach to predict severe OSA in children with loud snoring. Your comments and suggestions have substantially improved the quality of the study. Thank you very much!